# For 481 biomedical open access journals, articles are not searchable in the Directory of Open Access Journals nor in conventional biomedical databases

Mads Svane Liljekvist, Kristoffer Andresen, Hans-Christian Pommergaard and Jacob Rosenberg

Department of Surgery, Herlev Hospital, University of Copenhagen, Denmark

Corresponding author
Mads Svane Liljekvist,
m.liljekvist@gmail.com

## ABSTRACT

**Background.** Open access (OA) journals allows access to research papers free of charge to the reader. Traditionally, biomedical researchers use databases like MED-LINE and EMBASE to discover new advances. However, biomedical OA journals might not fulfill such databases' criteria, hindering dissemination. The Directory of Open Access Journals (DOAJ) is a database exclusively listing OA journals. The aim of this study was to investigate DOAJ's coverage of biomedical OA journals compared with the conventional biomedical databases.

**Methods.** Information on all journals listed in four conventional biomedical databases (MEDLINE, PubMed Central, EMBASE and SCOPUS) and DOAJ were gathered. Journals were included if they were (1) actively publishing, (2) full OA, (3) prospectively indexed in one or more database, and (4) of biomedical subject. Impact factor and journal language were also collected. DOAJ was compared with conventional databases regarding the proportion of journals covered, along with their impact factor and publishing language. The proportion of journals with articles indexed by DOAJ was determined.

**Results.** In total, 3,236 biomedical OA journals were included in the study. Of the included journals, 86.7% were listed in DOAJ. Combined, the conventional biomedical databases listed 75.0% of the journals; 18.7% in MEDLINE; 36.5% in PubMed Central; 51.5% in SCOPUS and 50.6% in EMBASE. Of the journals in DOAJ, 88.7% published in English and 20.6% had received impact factor for 2012 compared with 93.5% and 26.0%, respectively, for journals in the conventional biomedical databases. A subset of 51.1% and 48.5% of the journals in DOAJ had articles indexed from 2012 and 2013, respectively. Of journals exclusively listed in DOAJ, one journal had received an impact factor for 2012, and 59.6% of the journals had no content from 2013 indexed in DOAJ.

**Conclusions.** DOAJ is the most complete registry of biomedical OA journals compared with five conventional biomedical databases. However, DOAJ only indexes articles for half of the biomedical journals listed, making it an incomplete source for biomedical research papers in general.

## BACKGROUND

The idea of open access (OA) in the field of scientific research is to create a publishing platform where knowledge is freely available for all (*The Budapest Open Access Initiative, 2002*) and not bound by commercial interests (*Giglia, 2007*). From 1993 to 2009, the number of published OA articles increased from less than 250 to more than 191,000 articles a year. In 2008, OA articles constituted an estimated 20% of all scholarly articles published in that year (*Bjork et al., 2010*; *Laakso et al., 2011*). OA research papers can be deposited in online archives (repositories), published in OA journals or both (*Bjork et al., 2010*). OA research papers can be published with (most OA journals) or without being peer reviewed (e.g., private or institutional repositories and journals exercising post-publication peer review—i.e., F1000Research). The idea of peer reviewed OA journals combines free availability with the benefits of traditional, scholarly communication and editorial quality control through peer review processes.

In the field of biomedical research, both subscription-based and OA journals are listed in online databases such as MEDLINE and EMBASE as well as archived in online repositories such as PubMed Central (Table 1). These databases index articles published in the listed journals which can be searched via different online search engines, e.g., PubMed or Ovid. PubMed searches more than 23 million citations from journals primarily in MEDLINE or PubMed Central, as well as citations from journals in the U.S. National Library of Medicine's (NLM) catalogue and books from the National Center for Biotechnology Information (NCBI) Bookshelf (*U.S. National Library of Medicine, 2002*). Journal applications to a database are sorted and selected according to strict selection criteria (*Peña, Valero & Sicilia, 2004*; *U.S. National Library of Medicine, 1988*; *U.S. National Library of Medicine, 2014d*). The above-mentioned databases all evaluate subject, regularity and standard formal requirements, such as whether the applying periodical can be considered a scientific journal. However, some biomedical OA journals might presently be unable to comply with more specific selection criteria. For example, complying with technical demands like those set by PubMed Central (*U.S. National Library of Medicine, 2014d*) can be costly for small, independent journals. PubMed Central requires all full text articles to be submitted in specific Document Type Definition (DTD), eXtensible Markup Language (XML) format and tagged with the correct values for a number of identifiers (*U.S. National Library of Medicine, 2014c*). For evaluation of the journal, a sample package of selected articles is submitted in the XML format. Similarly, for indexing in MEDLINE, XML-tagged data for each article abstract and citation are required for submitted articles (*U.S. National Library of Medicine, 1990*). Current inclusion in MEDLINE automatically qualifies a journal for inclusion in PubMed Central (*U.S. National Library of Medicine, 2014d*). The NIH-chartered committee, the Literature Selection Technical Review Committee (LSTRC), determines selection for MEDLINE by evaluating the journals' editorial and production quality, quality of the journals' content as well as the journals' standing and contribution to their respective fields (*U.S. National Library of Medicine, 1988*). The NLM's Library Operations Division (LOD) determines selection for PubMed Central based on the Collection Development Manual of the National Library of Medicine

Liljekvist et al. (2015), *PeerJ*, DOI 10.7717/peerj.972

**Peerj**

**Table 1** Properties of the included databases.

| | DOAJ | MEDLINE | PubMed central | SCOPUS | EMBASE |
|---|---|---|---|---|---|
| Type and size of content | 9,700 journals | 5,600 journals | 2,100 journals | 34,000 journals and book series | 8,400 journals |
| Subjects | All scientific and scholarly | Biomedicine and clinical medicine | Biomedicine and clinical medicine | Health, life, social and physical sciences | Broad biomedicine, focus on pharmacology and clinical medicine |
| Journal quality control | Peer review | Peer review | Peer review | Peer review | Peer review |
| Can be searched by abstracts, authors and journal title | Yes | Yes | Yes | Yes | Yes |
| Specific, hierarchical topic search available | No | Yes (MeSH) | Yes (MeSH—not available for all entries) | Yes (MeSH and Emtree among others) | Yes (Emtree) |
| Availability of content | All content must be available online | Either available online or in print | Articles must be supplied for archiving | All content must be available online | Either available online or in print |
| Special requirements or topics of evaluation | Full and immediate **open access (OA)** to all of a journal's content **required**. Transparent OA policies, and editorial process | Evaluates standing and contribution | Evaluation based on the Collection Development Manual of the National Library of Medicine | Evaluated on journal policy, quality, standing, regularity and availability | Evaluated on scientific and editorial coverage |
| XML-submission | XML-submission of full-text articles optional | XML-submission of abstracts/citations required | XML-submission of full-text articles required | Database converts submission to XML | Database converts submission to XML |
| Required age before review for indexing | None, but must publish at least 5 articles per calendar year to stay indexed | Life span of at least 12 months and 40 published articles | 15–30 articles published, publisher dependent | Not explicitly required | Not explicitly required |
| Title and abstract in English required | No | Yes | Yes | Yes | Yes |
| Access cost | Free | Free | Free | Institutional subscription only | Institutional subscription only |
| Uses | Indexes and links to journals' homepages, along with providing journal metadata. Links to free full text articles, when article data has been provided by the journal | Links to full text articles, as well as free full text (if available) | Archives free full text articles from OA journals, and free articles from subscription journals under the NIH Grant Policy | Links to full text articles | Links to full text articles |
| References | *Directory of Open Access Journals, 2014a*; *Directory of Open Access Journals, 2014d*; *Directory of Open Access Journals, 2014f*; *Directory of Open Access Journals, 2015* | *Falagas et al., 2008*; *Peña, Valero & Sicilia, 2004*; *U.S. National Library of Medicine, 1988*; *U.S. National Library of Medicine, 1990* | *U.S. National Library of Medicine, 2014c* | *Falagas et al., 2008*; *Scopus, 2014*; *Elsevier, 2015* | *Embase, 2014*; *Peña, Valero & Sicilia, 2004*; *Elsevier, 2015* |

**Notes.**

DOAJ, Directory of Open Access Journals; MeSH, Medical Subject Headings; XML, eXtensible Markup Language.

(*U.S. National Library of Medicine, 2004*). PubMed Central includes OA journals not included in MEDLINE and can therefore be interpreted as more comprehensive than MEDLINE, when considering OA journals. However, the specific parameters of evaluation are not evident from *PubMed Central's Scientific Quality Standard* (*U.S. National Library of Medicine, 2014d*). EMBASE selects its journals based on scientific quality and editorial coverage. SCOPUS's Content Selection & Advisory Board (CSAB) selects journals by evaluating their journal policy, quality, standing, regularity and availability. Elsevier, owning both SCOPUS and EMBASE, converts submitted articles to a standardized XML DTD when received from the journal (*Elsevier, 2015*).

Strict inclusion criteria may create a barrier for newly established OA journals with regards to selection and indexing of their content in these biomedical databases and repositories, even if these journals publish high quality papers—hence making them hard to find for readers. This may create a barrier for readers, thus hindering timely and wide dissemination of research and compromising the purpose of OA (*Berlin Declaration on Open Access to Knowledge in the Sciences and Humanities, 2003*). The Directory of Open Access Journals (DOAJ) (*Directory of Open Access Journals, 2014c*) was founded in 2003 and, at the time of this study, listed more than 9,700 OA journals. Supplying articles for indexation is optional and currently more than 5,600 of the journals are searchable at the article level. DOAJ aims to cover all OA journals regardless of scientific subject (*Directory of Open Access Journals, 2014a*) and lists journals that target academic researchers by primarily publishing research papers (*Directory of Open Access Journals, 2014f*). For inclusion in DOAJ, full text papers must be made available to the readers in full and for free immediately upon publication (*Directory of Open Access Journals, 2014f*). Journals must have transparent access policies and exercise peer review (*Directory of Open Access Journals, 2014f*). However, DOAJ does not evaluate the journals' standing, contribution and whether their papers are "important" (*Directory of Open Access Journals, 2014f*). Therefore, DOAJ offers a window for newly established OA journals to get indexed in an online database—thereby facilitating the access to papers published in OA journals.

The purpose of this study was to investigate the distribution and overlap of biomedical OA journals between DOAJ and "conventional biomedical databases."

## METHODS

### Databases

In this study, we investigated the content of the OA-specific DOAJ and a number of databases and repositories that are not OA-specific. These databases and repositories are specified below and will be referred to as the "conventional biomedical databases". DOAJ is not regarded as "conventional" as it only indexes OA journals, unlike the rest of the included databases. In order to investigate the distribution of biomedical OA journals between DOAJ and the conventional biomedical databases, we retrieved journal lists from DOAJ and the following four conventional biomedical databases: MEDLINE, PubMed Central (PMC), EMBASE and SCOPUS (Table 1). Furthermore, data from the Journal Citation Reports (JCR) 2012 Science and Social Sciences edition were downloaded and

included. Data from the U.S. National Library of Medicine (NLM) journal catalogue was also included to identify journals found in PubMed, which were not indexed by either MEDLINE or PMC (*U.S. National Library of Medicine, 2002*). Data on activity, OA-status, publication language and 2012-impact factor were collected from the four databases and Journal Citation Reports.

## Data collection and inclusion criteria

Journal lists were freely available from the websites of DOAJ, EMBASE (including a listing of MEDLINEs journals), SCOPUS, PubMed Central and NLM and were retrieved in May 2014. Data from JCR was retrieved using institutional access via the University of Copenhagen in January 2014. Access to the content of EMBASE and SCOPUS are subject to an institutional subscription as well, even though a subscription is not necessary to access their journal lists.

Journals were identified either by their unique International Standard Serial Number (ISSN) or Electronic ISSN (EISSN). Journal records listed without either of these were excluded. All journals were cross-matched on ISSN and EISSN, so journals with ISSN incorrectly registered as EISSN (and vice versa), were correctly matched.

The dataset was constructed by merging the databases' journal lists, and aggregating data for matching journals. Figure 1 illustrates the process. From this comprehensive list a sample of journals was drawn following four inclusion criteria: Only journals that were (1) actively publishing, (2) releasing all content free of charge immediately upon publication (full and immediate OA), (3) prospectively indexed in one or more of the included conventional databases and/or DOAJ and (4) considered to be of biomedical subject were included in the study.

The study only included actively publishing journals. Since manual collection of information on the latest issue from every journal was deemed too labour intensive, the following database denotations for activity were used: For journals listed in SCOPUS, each was labelled as "active/inactive" in the journal list. No similar variable was available for journals in MEDLINE, EMBASE or PubMed, so in order to avoid underestimating the share of active OA journals found in these databases, all journals indexed in MEDLINE, EMBASE or PubMed Central were considered active. Exceptions were made for journals explicitly noted as inactive by an end publication year (MEDLINE, DOAJ) or a "predecessor"-status (PMC) (*Fogelman, 2009*) as long as this was not contested by information from one or more of the other databases. Where data on activity was collected manually, journals were considered active if they had published at least one article in 2013 or 2014.

For this study, only journals granting full and immediate access to all content were considered to be OA journals in accordance with the *Bethesda Statement on Open Access Publishing (2003)* and the DOAJ selection criteria (*Directory of Open Access Journals, 2014f*). Subscription journals with optional OA for individual articles (hybrid OA), subscription journals allowing the authors to archive free versions of individual articles, journals providing OA to only part of their contents (e.g., research articles only) and

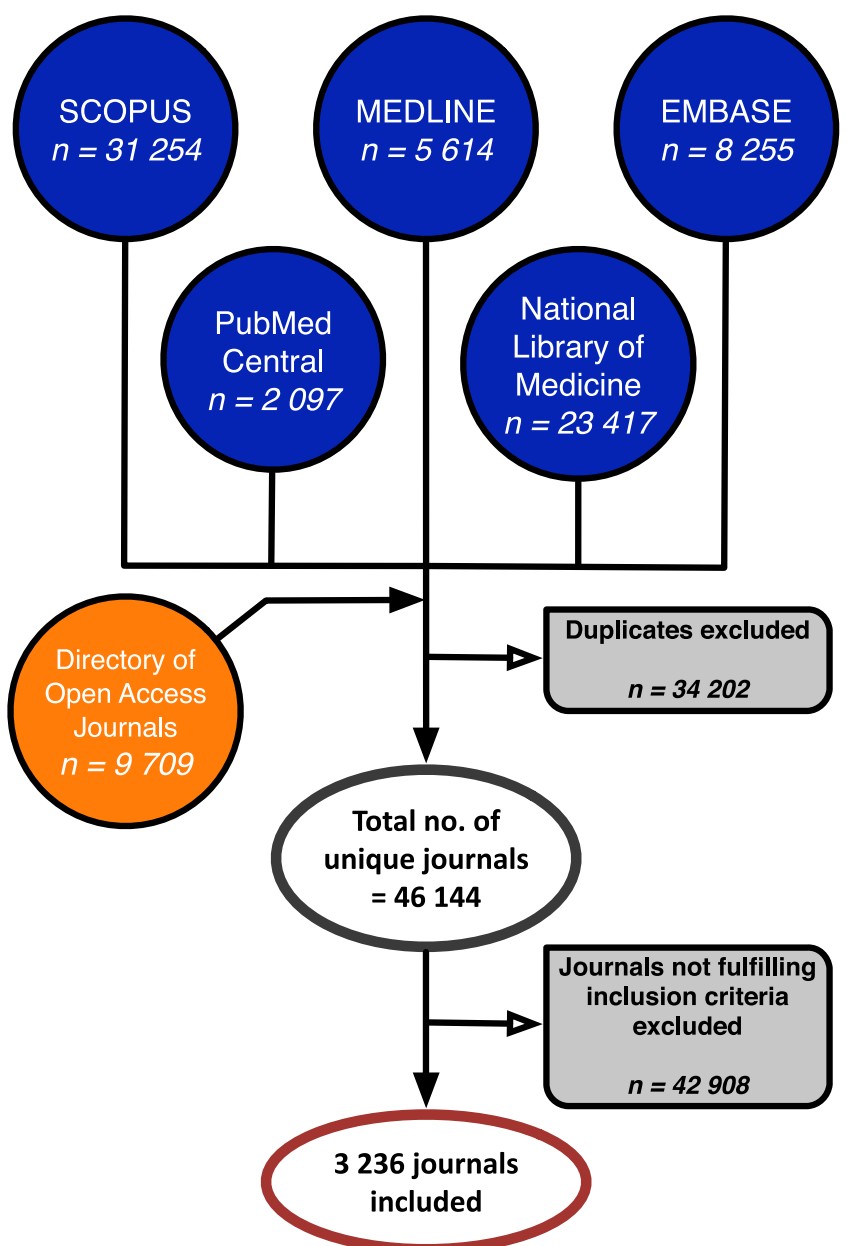

**Figure 1** The inclusion process of biomedical open access journals.

journals providing OA to their content after an embargo period (delayed OA) were not considered full OA for the purposes of the present study. SCOPUS, DOAJ and PubMed Central provided information on OA-status. If any one of these databases had labelled the journal as OA, the journal was included. For 403 actively publishing journals, OA-status could not be determined via data from the downloaded journal lists. OA-statuses for these journals were collected manually using the journal websites.

Since the journal must be both currently publishing and grant full and immediate OA to all content to be eligible for inclusion in DOAJ (*Directory of Open Access Journals, 2014f*), all

journals listed in DOAJ were assumed to fulfil these two criteria—except when an end year of publication (when the journal had ceased to publish) was listed in DOAJ.

MEDLINE, PubMed Central, SCOPUS and EMBASE index the contents of their selected journals prospectively. DOAJ indexes articles prospectively when supplied by the journal. Therefore, all journals listed in DOAJ were considered as being prospectively indexed. NLM's catalogue lists titles from all of PubMed—including MEDLINE's and PMC's repertoires as the active sources for new citations. Furthermore, it lists titles no longer being indexed along with non-biomedical titles etc. (*U.S. National Library of Medicine, 2002*). Therefore, titles exclusively listed in NLM's catalogue were not considered prospectively indexed in any of the included databases.

Both DOAJ and SCOPUS select journals from a broad spectrum of scientific fields. Only journals of a biomedical subject were included in this study. The chosen biomedical subjects from DOAJ and SCOPUS are presented in Table S1 and S2, respectively. Journals indexed in EMBASE, MEDLINE or PubMed Central were considered biomedical since these databases only index journals of biomedical subject (*Embase, 2014*; *U.S. National Library of Medicine, 1988*; *U.S. National Library of Medicine., 2014b*).

Of the included journals, 552 were not in DOAJ. For these journals, data on activity and OA-status were collected manually via their respective websites. Journals found inactive or not full OA were excluded. In total, 434 journals were left after exclusion of erroneously included journals.

Furthermore, 283 journals had no language information available through the databases. Languages for these journals were collected manually.

Seventy-three included journals had conflicting information on activity (5 with an end publication date, 10 denoted as "predecessor" and 58 denoted as "inactive" by SCOPUS). These were manually checked for activity and OA-status. Nine of these journals were inactive, and were excluded. All of the journals were full OA.

To determine how many of the journals indexed in DOAJ had opted to submit the metadata of their contents to DOAJ, we downloaded article metadata from DOAJ's XML-based metadata server (*Directory of Open Access Journals, 2014e*). We used a modified OAI-PMH (Open Access Initiative Protocol for Metadata Harvest) (*Lagoze et al., 2008*) C# client for scraping the required article metadata (Table S3). If any article indexed in DOAJ had a publication date in 2012 or 2013 and carried the journal's ISSN, that journal would be marked as having their content the respective year indexed in DOAJ.

### Data validation

To validate the data, SPSS's "Random Sample" function was used to draw two samples from both the excluded and the included journals followed by manual collection of specific data for the sampled journals. This was done to (a) make sure we had not wrongfully excluded relevant journals, and (b) to ensure the reliability of data for the included journals.

To check for wrongfully excluded journals, 100 journals excluded for being inactive or subscription based were randomly sampled. Activity and OA-status were manually collected from the NLM catalogue and the journals' respective websites. The access level of

any active journal was determined based on the availability of both current issue contents and archived content.

Another random sample of 160 (∼5%) included journals was drawn to verify activity, OA-status, language category, impact factor and whether the journal was indexed in DOAJ. The data was manually collected from the journal websites, JCR and DOAJ.

### Data presentation

Primarily, the distribution and overlap of OA journals between the conventional biomedical databases and DOAJ was determined. All analyses were carried out with SPSS 22 software (IBM Corporation, Armonk, New York, USA). Continuous data (not normally distributed) are reported as median [interquartile range]. Categorical data are reported in percentages (numbers).

## RESULTS

### Data validation

The sample of 100 excluded journals yielded 57 active journals; including 1 full OA journal (*Oklahoma Law Review*), which was not of biomedical subject even though it was categorized under '*Medicine*' in SCOPUS. Overall, none of the sample journals fulfilled the inclusion criteria.

The sample of 160 (∼5%) included journals yielded the following:

- All examined journals were active, full OA-journals except for 1 journal (*BMC Pharmacology*), which had fused with *BMC Clinical Pharmacology* (which had been rightly excluded) in 2012 to form another new journal. This new OA-journal was already included in our cohort and so the predecessor was removed from the cohort.
- All journals were correctly labelled as "not in DOAJ" ($n = 33$), or "in DOAJ" ($n = 126$).
- For 5 journals, language had been collected manually, since no information was available through the databases. Of the remaining 154 journals, 3 had been incorrectly labelled regarding English/non-English language.
- All journals with a 2012 impact factor ($n = 33$) had the correct impact factor assigned during the dataset build. The remaining 126 journals had correctly been assigned no 2012 impact factor.

### Findings

In total, 3,236 biomedical OA journals were included in this study. Of these, 89.2% (2,888 journals) were published in English and 19.5% (632 journals) had received an impact factor for 2012 with a median value of 1.257, interquartile range (IQR): [0.615–2.423]. The overall journal distribution by database type (DOAJ or conventional), language (English/non-English) and impact factor is shown in Table 2.

The proportions of the included OA journals listed in the respective databases are summarized in Table 3. We found that 86.7% (2,804 journals) of the included OA journals were listed in DOAJ. In contrast, each of the conventional databases accounted for lesser proportions of the study sample. Combined, the conventional biomedical databases listed

**Table 2** The distribution and overall characteristics of biomedical open access journals between the Directory of Open Access Journals and the conventional biomedical databases.

| | In DOAJ | Only in DOAJ | In both DOAJ and conventional biomedical databases | Only in conventional biomedical databases | In conventional biomedical databases | All open access journals |
|---|---|---|---|---|---|---|
| Number of biomedical OA journals | 86.7 (2,804) | 24.9 (807) | 61.7 (1,997) | 13.3 (432) | 75.0 (2,429) | **100 (3,236)** |
| English language journals | 88.7 (2,488) | 76.5 (617) | 93.7 (1,871) | 92.6 (400) | 93.5 (2,271) | **89.2 (2,888)** |
| Received impact factor 2012 | 20.6 (579) | 0.1 (1) | 28.9 (578) | 12.3 (53) | 26.0 (631) | **19.5 (632)** |
| Impact factor 2012 (median [interquartile range]) | 1.316 [0.619–2.456] | 0.372 [0.372-0.372] | 1.320 [0.619–2.458] | 0.994 [0.558–1.892] | 1.263 [0.615–2.426] | **1.257 [0.615–2.423]** |

**Notes.**
Values are presented as % (n), if nothing else is noted.
Percentages in the first row are based on total number of journals included.
Percentages in all other rows are based on the number in journals in each group (i.e., the (n) of each columns first row).
DOAJ, Directory of Open Access Journals; OA, open access.
Conventional biomedical databases include: MEDLINE, PubMed Central, EMBASE, SCOPUS and U.S. National Library of Medicine.

**Table 3** The distribution of biomedical open access journals among the included databases.

| Database | Journals indexed | Journals unique to each database | Journals not found in DOAJ |
|---|---|---|---|
| DOAJ | 86.7 (2,804) | 24.9 (807) | – |
| PubMed (search engine) | 56.4 (1,824) | 0 (0) | 9.0 (292) |
| - MEDLINE | 18.7 (605) | 0 (0) | 2.7 (88) |
| - PubMed Central | 36.5 (1,181) | 0 (0) | 5.4 (176) |
| - Rest of PubMed[a] | 10.3 (334) | – | 1.9 (60) |
| SCOPUS | 51.5 (1,667) | 0.6 (19) | 6.7 (217) |
| EMBASE | 50.6 (1,636) | 2.6 (83) | 9.6 (312) |

**Notes.**
Values are presented as % (n).
DOAJ, Directory of Open Access Journals; OA, open access.
[a] Journals only searchable via PubMed through the National Library of Medicine's catalogue, but not prospectively indexed via MEDLINE or PubMed Central. They are included as they are prospectively indexed in one or more of the other databases.

75.1% (2,429 journals) of the included journals—18.7% (605 journals) in MEDLINE, 36.5% (1,181 journals) in PubMed Central, 51.5% (1,667 journals) in SCOPUS and 50.6% (1,636 journals) in EMBASE. Using PubMed to search for content in OA journals (displaying results from MEDLINE, PubMed Central and NLM's catalogue combined) revealed results from a large proportion of OA journals as 56.4% (1,824 journals) are listed in these three databases combined. However, 10.3% (334 journals) were not prospectively indexed in MEDLINE or PMC. These titles were listed in the NLM's catalogue and therefore did not have full or current content searchable via PubMed. Hence, only 46.1% (1,490 journals) of the biomedical OA journals are being prospectively indexed in full for PubMed via MEDLINE and PubMed Central. Considering journals only listed in one

**Table 4** **The overlap of biomedical open access journals between all included databases.**

| | The number of journals from the 1st row listed database | | | | |
| --- | --- | --- | --- | --- | --- |
| | Database name | DOAJ | EMBASE | SCOPUS | MEDLINE | PMC |
| Which can be found in the 1st column listed database | DOAJ | 100.0 (2,804) | 80.9 (1,324) | 87.0 (1,450) | 85.5 (517) | 85.1 (1,005) |
| | EMBASE | 47.2 (1,324) | 100.0 (1,636) | 74.4 (1,241) | 99.3 (601) | 67.1 (793) |
| | SCOPUS | 51.7 (1,450) | 75.9 (1,241) | 100.0 (1,667) | 92.1 (557) | 66.9 (790) |
| | MEDLINE | 18.4 (517) | 36.7 (601) | 33.4 (557) | 100.0 (605) | 24.9 (294) |
| | PMC | 35.8 (1,005) | 48.5 (793) | 47.4 (790) | 48.6 (294) | 100.0 (1,181) |

**Notes.**

Values are presented as % (n).

DOAJ, Directory of Open Access Journals; PMC, PubMed Central.

The table should be read with 1st row representing the databases of origin and 1st column as the crosschecked databases. Thus, of journals from a 1st row database, % (n) can be found in the crosschecked 1st column database.

database, the conventional biomedical databases had 3.2% (102 journals) uniquely listed in one of the databases (Table 3) compared to 24.9% (807 journals) only in DOAJ.

The overlap of journals with all of the included databases in between is illustrated in Table 4. DOAJ lists between 80.9% (1,324 journals) to 87.0% (1,450 journals) of the biomedical OA journals found in each of the conventional biomedical databases. Inversely, the conventional biomedical databases overlapped DOAJ with between 18.4% (517 journals) and 51.7% (1,450 journals).

Publishing in English was common for both the journals listed in DOAJ (88.7%, 2,488 journals) and the journals listed in the conventional biomedical databases (93.5%, 2,271 journals) (Table 2). However, of the 807 journals listed only in DOAJ, a smaller proportion of 76.5% (617 journals) were published in English. Meanwhile, 92.6% (400 journals) of the 432 journals listed only in the conventional biomedical databases were published in English.

Considering journal impact factor (Table 2), 20.6% (579 journals) of the journals listed in DOAJ had received an impact factor for 2012 with a median value of 1.316, IQR: [0.619–2.456]. For journals listed in the conventional biomedical databases, 26.0% (631 journals) had received a 2012 impact factor with a median value of 1.263, IQR: [0.615–2.426]. Journals only listed in DOAJ and journals listed only in the conventional biomedical databases had only 0.1% (1 journal) and 12.3% (53 journals) respectively with an impact factor for 2012. The impact factors were 0.372 and median 0.994, IQR: [0.558–1.892] respectively.

Some of the journals listed in DOAJ had articles from 2012 (51.1%, 1,434 journals) and 2013 (48.5%, 1,359 journals) indexed in DOAJ (Table 5). Of the journals listed only in DOAJ, 40.5% (327 journals) and 40.4% (326 journals) had articles from 2012 and 2013 respectively indexed in DOAJ.

Twenty journals only listed in DOAJ and with no content from 2013 were randomly selected (via SPSS) and their method of dissemination further scrutinized. Of these, 90% (18 journals) had some content accessible via Google Scholar. Fifty-five per cent (11 journals) had no other databases listed on their homepages. The remaining 45%

**Table 5** The proportion of biomedical open access journals listed in the Directory of Open Access Journals that has their content indexed at article level.

|  | Articles published in 2012 | Articles published in 2013 |
| --- | --- | --- |
| Share of journals in DOAJ with indexed articles | 51.1 (1,434) | 48.5 (1,359) |
| Share of the journals *only found* in DOAJ and with articles indexed herein | 40.5 (327) | 40.4 (326) |

**Notes.**
Values are presented as % (n).
DOAJ, Directory of Open Access Journals; OA, open access.

(9 journals) listed databases including CrossRef, Index Copernicus, SciELO, Redalyc, LatIndex, Open J-Gate and JournalSeek. All 20 journals had content available through their homepage or their publisher's homepage.

## DISCUSSION

### Main findings

This study found that DOAJ lists the vast majority of biomedical OA journals and overlaps each of the conventional biomedical databases in an equal manner and also includes a number of journals not listed in the conventional biomedical databases. Even combined, MEDLINE, PMC, SCOPUS and EMBASE did not match DOAJ's number of listed biomedical OA journals. Each of the conventional biomedical databases listed about half of the journals relevant to this study. However, DOAJ alone did not list all biomedical OA journals, leaving 13.3% of to be located elsewhere.

Both the journal subset not listed in DOAJ and the journal subset not listed in the conventional biomedical databases were characterized by fewer journals with a 2012 impact factor and a lower median impact factor value. Only one journal outside the conventional biomedical databases had received an impact factor for 2012. To receive an impact factor, a journal must be selected for and indexed in Web of Science (WoS) where citation counts are published as impact factor via Journal Citation Reports. Before being selected for WoS, a journal is evaluated on a range of parameters similar to the conventional biomedical databases' selection criteria. These include timeliness of publication, keeping with international editorial conventions, available bibliographical information in English and peer review processes (*Testa, 2012*). Language was dichotomized into English/non-English and the majority of the journals listed in both the conventional biomedical databases and DOAJ were published in English. However, the lowest percentage of English-publishing journals (76.5%) was found among journals not listed in the conventional biomedical databases but only in DOAJ. Amongst journals listed in the conventional biomedical databases, this percentage was higher (93.5%) indicating that English-publishing journals might be favoured.

### Strengths and limitations

This study focused on a single scientific field to keep "cultural" differences between the various scientific disciplines from confounding the overall picture. Several large databases were included in this study and hence a large number of potentially relevant journals

were screened for inclusion in the study. Four relevant inclusion criteria were applied to define the cohort from database metadata. Followingly, the cohort was refined through sequential manual exclusion of ineligible journals, which had been wrongfully included. The systematic inclusion of journals was based on an assumption that the database data was correct. It was assumed that all journals in DOAJ were both active and full OA. The activity of OA journals in DOAJ has earlier been contested (*Morris, 2006*), where up to 14% of journals were found not to be currently active. DOAJ has changed a lot since 2005 and currently uses a standardized application form (*Directory of Open Access Journals, 2014d*) along with running exclusion of inactive journals (*Directory of Open Access Journals, 2014b*). To validate the assumption, and because the journals' metadata derived from all databases could be faulty, a limited data validation of both included and excluded journals was conducted. This revealed high concordance between database data and manually collected data, ensuring that only ineligible journals had been excluded and only eligible journals had been included. One limitation of this study is that exclusively full OA journals were included—e.g., excluding journals providing OA to scientific content only, journals exercising delayed OA along with journals employing hybrid OA business models. One could argue that inclusion of these journals would alter the results since these business models do not comply with the DOAJ selection criteria (*Directory of Open Access Journals, 2014f*). Thus these journals would not contribute to the segment found in DOAJ. The study found that the majority of the included journals were published in English. However, specific non-English platforms such as SciELO, Redalyc and Latindex were not included. These platforms index OA journals from Latin America, the Caribbean, Portugal and Spain and might contain even more biomedical OA journals not publishing in English.

## Perspectives

To our knowledge, this is the first study directly comparing DOAJ's amount of listed journals with that of conventional biomedical databases. Earlier studies of OA publishing have utilized DOAJ as an assumed complete list of OA journals, and drawn their samples from here (*Dallmeier-Tiessen et al., 2010*; *Laakso & Bjork, 2012*; *Laakso et al., 2011*). Other studies compared the attributes of some conventional biomedical databases (*Falagas et al., 2008*; *Kejariwal & Mahawar, 2012*), but not the exact overlap of listed journals.

Medicine is one of the scientific fields previously shown to rely on OA journals rather than self-archiving for distributing OA content (*Bjork et al., 2010*). PubMed is the primary search engine for many biomedical researchers, making the content of thousands of journals searchable, including approximately 1,800 OA journals. To display OA articles only, the user can enable the "Free Full Text"-filter when searching PubMed (*U.S. National Library of Medicine, 2014a*). However, this function also displays OA papers from otherwise subscription journals (hybrid OA journals) and presupposes the PubMed indexer designates all relevant articles correctly. Similarly, DOAJ can be searched at article level using Boolean operators. A major condition for considering DOAJ equal to the conventional biomedical databases is DOAJ's indexation of individual journal articles in such a fashion so that they become searchable for the readers. However, we found

that only about 50% of the biomedical journals had actually opted to get their articles indexed in DOAJ. This is an important fact to consider, as it means DOAJ's coverage at the article level is lacking compared to the databases where article indexation is a main feature—e.g., MEDLINE and PMC. With 807 biomedical OA journals not reachable via the conventional biomedical databases but readily found through DOAJ, we may raise the question whether searches in DOAJ should be included along the conventional PubMed and EMBASE searches when conducting systematic reviews of the biomedical literature. However, as only about 40.5% of these journals have current content available, which is searchable through DOAJ, the actual gain from searching DOAJ for individual articles could be limited. It does seem odd that journals exclusive to DOAJ would be reluctant from having their content be made available and searchable. Reasons for this lack of indexed articles for these journals are only speculative at this stage. Thus, it would be interesting to search out and assess the number of conducted studies that are only found via DOAJ. This would shed light on the size of includable articles left out of the systematic reviews that do not search DOAJ. However, such an assessment is not within the scope of the present study. We found that the majority of journals without content in DOAJ or the conventional biomedical databases have some content searchable via Google Scholar. However, to what degree is uncertain, and the completeness of Google Scholar's searches have been contested (*Kejariwal & Mahawar, 2012*). Nearly half of the journals had no other databases listed on their websites, and thereby Google Scholar might supposedly be the only search engine, besides the journals' respective archives, where readers can find their content.

The online databases discussed in this study aggregate data on journals and their articles as a service to readers. This makes them attractive options for journals to be listed in. However, to get indexed in a database, journals must apply to the specific database and make sure the journal fulfils the database's pre-specified selection criteria. For the conventional biomedical databases included in this study, selection criteria include the quality of content, production and home pages, along with the editorial work, the quality of peer review and evaluation of a journal's academic standing and contribution to its field (*Peña, Valero & Sicilia, 2004*; *Scopus, 2014*; *U.S. National Library of Medicine, 1988*; *U.S. National Library of Medicine, 2014d*). Furthermore, journals applying to e.g., MEDLINE must have a minimum number of papers published and comply with the specific technical requirements mentioned earlier (Table 1) (*U.S. National Library of Medicine, 1988*). These criteria are set in order to secure the user a certain level of scientific quality within the included journals and their papers, but the criteria compromise the extent to which journals can be included. DOAJ's selection criteria do not depend on evaluation of e.g., standing, contribution and layout in the same way criteria for some of the conventional biomedical databases do. DOAJ demonstrates a database model where biomedical OA journals not presently selected for the conventional biomedical databases can still have their content indexed and made available and searchable for readers, aiding wide dissemination of their content. However, of the journals in DOAJ 48.5% (1,359 journals) had articles from 2013 indexed in DOAJ. This leaves 51.5% (1,445 journals) with no current articles indexed in DOAJ. This lack of journals indexing their articles might

have root in the fact that DOAJ requires either a manual (article by article) or XML-coded submission of article metadata to index (*Directory of Open Access Journals, 2015*). This complicated and/or expensive workload might keep journals from heeding DOAJ's strong recommendation to upload article metadata. Furthermore, for 432 biomedical OA journals, no record in DOAJ could be found even though they are listed in one or more of the conventional biomedical databases. One can only speculate as to why these journals are not listed in DOAJ. Their publishers might be content with the attention gained from readers through the conventional biomedical databases, they might find applying for selection for DOAJ unnecessary, or they might be oblivious to DOAJ's existence.

## CONCLUSIONS

The Directory of Open Access Journals lists the majority of biomedical OA journals. It also overlaps and lists more than 80% of each of the conventional biomedical databases' biomedical OA journal, and has the largest proportion of uniquely listed biomedical OA journals among the studied databases. However, less than half of the listed journals have current content indexed in DOAJ. The conventional biomedical databases each lack around 50% of relevant biomedical OA journals and their inclusion of the OA journals listed in DOAJ is sparse and unevenly distributed among the databases.

### Funding
The authors declare there was no funding for this work.

### Competing Interests
The authors declare there are no competing interests.

### Author Contributions
- Mads Svane Liljekvist conceived and designed the experiments, performed the experiments, analyzed the data, contributed reagents/materials/analysis tools, wrote the paper, prepared figures and/or tables, reviewed drafts of the paper.
- Kristoffer Andresen conceived and designed the experiments, performed the experiments, analyzed the data, reviewed drafts of the paper.
- Hans-Christian Pommergaard conceived and designed the experiments, analyzed the data, reviewed drafts of the paper.
- Jacob Rosenberg conceived and designed the experiments, reviewed drafts of the paper.

### Supplemental Information
Supplemental information for this article can be found online at http://dx.doi.org/10.7717/peerj.972#supplemental-information.

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
