# Peer review of "For 481 biomedical open access journals, articles are not searchable in the Directory of Open Access Journals nor in conventional biomedical databases"

_PeerJ, doi:10.7717/peerj.972_

## Round 0.1 · original submission · Major Revisions

Please pay particular attention to the comments of Reviewer 2

·

Basic reporting

The manuscript is well written, the arguments clearly explained with all necessary details. The study represents a useful contribution to knowledge and it is critical about strengths and limitations.
The structure conforms to acceptable format and all figures are relevant. Supplementary material is useful.
The text is subdivided in coherent sections. Data validation is appreciated.
Suggestions for improvement
1) Specify the attributes of “conventional” databases and indicate from the very beginning why the DOAJ is not “conventional”. If one of the attributes is the availability of hierarchical topic search (such as MeSH) , make it clear that this is not available for all entries in PubMed Central
2) Lines 22-23. The purpose of OA is not only “timely dissemination of research”, but also wide dissemination exempt of any kind of access barriers (see for example the Berlin Declaration)
3) Access to EMBASE and SCOPUS is subject to an institutional subscription; this should be made clear in the text of the article and not only in the Table (Table 1), since it is a basic feature for readers.
4) Mention of other databases of OA journals such as SciELO, Redalyc and Latindex, including mostly Latin American Journals (mostly in Spanish) may provide additional comments, as regards language of OA journals, and peripheral OA journals (possibly to be included in the limitation section).
Corrections
1) Line 145: 19.5% and not 19,5%
2) Line 150: 75%, include decimal point and one digit
3) Line 230: 40%, include decimal point and one digit
4) Figure 1. National Library of Medicine (not National Libraries of Medicine)

Experimental design

The investigation was conducted rigorously and it is described with sufficient information so as to be reproducible.

Validity of the findings

Data are correct, and controlled. Conclusions are connected with the original question and consistent with the title of the article.

Additional comments

The article meets the Peer J criteria. Suggestions provided can improve it.

·

Basic reporting

The article includes sufficient introduction, however I would recommend describing the differences in the indexing of articles between the databases described in the research in the int. Furthermore, I recommend mentioning the existence of OA without peer review.

Experimental design

Due to the nature of the paper, I believe additional information and research should be conducted. Researchers should provide a table with overlap of all journals in the databases studies, as well as the number of journals only available in DOAJ, but that also have articles indexed. Furthermore as in the discussion thy mention that Sys reviews might be affected, searching at least the OA whose articles are only indexed in DOAJ and not in conventional databases, could provided an insight in the number of RCTs that are published in those journals. Also as an appendix a database with all listed journals and their indexation in the databases would be appreciated.

Validity of the findings

Please see the general comments for recommendations on how to improve your discussion, and the phrases used.

Additional comments

Background:

Line 15 – 16 – All OA journals have peer review? Can you please mention examples of non peer review OAs, or those with post publication peer review? Please also mention that DOAJ includes only OA with peer review.

Line 18 – I would not call both PubMed central and Medline databases, as one a repository of full articles and the other of citations.

Lines 18-19. Please mention the inclusion criteria for PubMed central and how that differs from Medline. I would also strongly advise mentioning the differences in the indexing processes, especially the indexing that is dependent upon submission of data or xml by the journals to the databases versus the bases doing this work for the journals.

Line 28, no need to repeat „which is searchable at article level and freely available to researchers“ this has already been mentioned in the introduction.

Line 44 - and were retrieved IN May 2014.

Lines 56-59, Is there a need for commas after brackets: “1),”

Data validation

Line 119 and 122 - Please describe the random sampling methods used to obtain these journals.

Line 130 - Is there a need for both IQR and full range? Line 156 reports only range?

Findings:

Line 145 - change 19,5% to 19.5%

I would advise authors be consistent in their reporting, either always report % followed by (n), or everywhere just report the percentages.

Line 152, PubMed is a search engine not a database, therefore reformulate the sentence. Also in Table 3

Line 153 – I would advise putting this sentence, after the third one, and not having it as the first sentence of your findings, as the first results should be the total number of OA journals you found.
Line – As you also looked at scopus, do you have SCIImago journal ranks? Furthermoe, these 19.5% can u mention are they indexed in all the database you searched?

Table 2 – Can you please add the totals, - shouldn’t the percentages always be based on the total 3,236 journals. Please indicate otherwise.


Line 157 - The proportions of journals in the respective databases – this sounds as proportion of OA journals compared to all other journals in that database.

Line 158 – Strange wording, as you first collected all of them from DOAJ anyway. “We found, that 86.7% (2,804 journals) of the included OA journals could be found”

Lines 159-162, it would be more interesting to report unique OA in these databases

Line 162 – Again Pubmed is not a database

Line 166 - current content searchable through PubMed

Lne 167 – What about other databases searchable through PubMed

Lines 173-180 – I would advise stat. Testing to see the differences between these groups, if that is the point of this paragraph, although IF is obtained only for those in JCR, so other impact measurements would be needed.

Line 179 – groups? Isn’t one groups made of only 1 journal?

Line 181– Which subset? Please rephrase and make more clear.

Discussion:

Line 176 – Please rephrase, this is almost self evident from your introduction, as the criteria for inclusion differs between DOAJ and „conventional“ databases

Line 179-181 – Please mention that it is the journals that apply to these databases, it is not the responsibility of the databases to search for journals or added them on their own accord.

Line 182 – Please rephrase, you obtained the full number of OA journlas only by looking at the (lack of) overlap of journals in these databases, not by searching for journals through other means. Also you did not include information on the year when the first journals started, as that could be one of the reasosn they were not included in the databases.

Line 185-186, this is unfounded, as the criteria for receving IF is incluson in WoS, and depends upon their selection.

Line 220 - Free Full Text filter does not include only full OA journals, it can also list articles that are OA on otherwise subscription based journals. Furthermore some articles that are OA can also lack such designation if its not provided by the indexer.

Lines 241-250 this should be in the introduction, please rewrite this paragraph

Line 251 – subjective – please find another term, DOAJ demands peer review and includes only OA, why is that less subjective?


Line 258-260 this is not conclusion of your work, but discussion, please delete, and focus only on your results or their implications.


Figure 1 – National LibrarieS – plural?

Table 1 – Please be consistent with dots. Add dots after yes in row 4, similar to No in fifth row. Also is there a need for brackets in the last row, and again add dots at end.
Is there a need for figure 1 if you include data from JCR and NLM in this?

Table 3 - Can you please provide as an additional table or extra rows/columns, no of unique journals indexed in each of the conventional databases – how many OA journals are only in scopus, medline and etc? And what are the overlaps between these databases?

Please discuss possible reasons why journals do not index their articles in DOAJ.

Additional questions and observations for the researchers:
Did u obtain the average number of articles published by a journal in a year?
Based on your findings did you communicate the discrepancies you found to the major databases so they correct their records?

Reviewer 3 ·

Basic reporting

The abstract is written clearly, concisely and legibly, and refers to the subject of the manuscript.
The introduction states the main problem of the study and a concise presentation of basic knowledge, and then the aim of this study – investigate the distribution and overlap of biomedical OA journals between DOAJ and conventional biomedical databases. I recommend further describing selection criteria for indexing articles in conventional biomedical databases compared to DOAJ, so readers better understand why is so many more journals indexed in DOAJ then in conventional biomedical databases.
The section relating to methods of research contains sufficient details.
The results are presented with sufficient details and generally correspond to the aim of research and includes data validation and findings. In the discussion authors present the main findings, especially those arising from the statistical results.
The figures complement the text. Their quality is suitable for publication and understandable. I think that literature used very well covers the subject of research.
Finally, as the authors well placed the topic of research, explored the available literature, conducted quality methodology research, and based on their research properly presented their views and conclusion.

Experimental design

No comment

Validity of the findings

No comment

Additional comments

I would only recommend to author to further describe selection criteria for indexing articles in conventional biomedical databases compared to DOAJ, so readers better understand why is so many more journals indexed in DOAJ then in conventional biomedical databases.

---

## Round 0.2 · Minor Revisions

Please address the comments of reviewers, especially those of reviewer No. 2. Style comments can be disregarded.

·

Basic reporting

The manuscript has improved a lot after the first revision, taking into account the comments by the three reviewers which many points in common.
The points I have raised in the first review were addressed in the manuscript.
Still to be revised:
1) In the reference list, the citation to the Berlin Declaration of OA. The concept has been included in the text (as suggested) , but reference is not correctly cited in the reference list. In fact, it is listed under the name of Stratmann (2003) as author of the Declaration, while Stratmann is only the contact person to receive information on the Declaration, not the author. The reference therefore should be listed under the title “Berlin Declaration….”

Experimental design

Revision much improved the text

Validity of the findings

Data are correct. Conclusions are connected with the original question and consistent with the title of the article.

Additional comments

The article meets the Peer J criteria. Review improved the report.Only revise one reference in the reference list.

·

Basic reporting

Would advise stressing the difference between indexing of articles and listing of journals. Furthermore discussion section could benefit with a bit more citations from suggestions and observations of other authors in the field.

Experimental design

no comments

Validity of the findings

Please check below in the general comments.

Additional comments

Title:
I would advise changing the title to: Not even Doaj covers all of the open access biomedical journals or:
For almost 500 OA biomedical journals articles are not searchable in DOAJ nor in conventional databases.
(As I understand its 807-326 =481)
Abstract:
Disseminate? Perhaps better word would be allow free access
The Directory of Open Access Journals (DOAJ) is a database searchable at article level – yes and no, in what percentage is stated later, so would rephrase this here.

Methods – are not consistent with methods section in the article were it states that 4 conventional databases were searched. And here 5 are listed, but only 4 percentages in the results (please correct this also in the table).

Line 16. OA articles can be published with or without (e.g. F1000Research) being peer reviewed.

Line 17 – can you provide an example of a repository that reviews OA papers before allowing them in? Don’t they do this once to review the actual journal?

Article
Data presentation:
Line 172 – are commas needed behind median and percentages?

Discussion:
Main findings
Line 235-238 I would highly advise using another term instead of coverage as it does not specify indexing of journals, or articles. As researchers search for journal articles, does that mean that more coverage of articles is available in DOAJ? As you claim only 1359 of journals are searchable by article level in DOAJ, do not the conventional databases therefore cover more articles, while DOAJ indexes more journals?

Lines 245-246 please delete or rephrase and mention what is needed for a journal to receive an IF factor, and where it needs to be indexed.

LINE 247 – Simplify this sentence. Majority of journals in both the conventional databases and DOAJ are published in English.

Line 250 Is there any language restriction for inclusion in conventional databases? Please do not make this assumption, could it not be we are seeing a new rise of local journals? Without data on the origin of journals these claims are unfounded. I would actually advise deleting sentences 248-251 as you comment them better in 273-276, which could then be moved here instead of 248-251.

Discussion
I would like authors to comment on the fact that some OA journals that are exclusive to DOAJ still do not submit their articles in there – could authors check a few of these and see where their articles are indexed (or are they only online on the journals website). Also could authors comment on possible reasons why OA journals indexed in conventional databases are not indexed in DOAJ. Could a list be forwarded to DOAJ and have them actively approach the journals?

On the last note – quality of English language is not equal throughout the article, would advise the authors to have a native speaker help them address the flow of the text.

---

## Round 0.3 · accepted · Accept

Thank you for addressing all reviewers' comments.